# Fibromyalgia and Depression in Women: An 1H-NMR Metabolomic Study

**DOI:** 10.3390/metabo11070429

**Published:** 2021-06-30

**Authors:** Carmen Marino, Manuela Grimaldi, Paola Sabatini, Patrizia Amato, Arianna Pallavicino, Carmen Ricciardelli, Anna Maria D’Ursi

**Affiliations:** 1Department of Pharmacy, University of Salerno, Via Giovanni Paolo II, 84084 Fisciano, Italy; cmarino@unisa.it (C.M.); magrimaldi@unisa.it (M.G.); ariannapallavicino@gmail.com (A.P.); carmen.ricciardelli@gmail.com (C.R.); 2U.O.C. Clinical Pathology D.E.A. III Umberto I, Viale S. Francesco D’Assisi, 84014 Nocera Inferiore, Italy; paola.sabatini1@gmail.com; 3ASL Ser. T Cava de’ Tirreni, Piazza Matteo Galdi 1/3, 84013 Pregiato, Italy; patriziaamato277@gmail.com

**Keywords:** fibromyalgia syndrome, NMR metabolomics, Hamilton test

## Abstract

Fibromyalgia is a chronic and systemic syndrome characterized by muscle, bone, and joint pain. It is a gender-specific condition with a 9:1 incidence ratio between women and men. Fibromyalgia is frequently associated with psychic disorders affecting the cognitive and emotional spheres. In the reported work, we compared 31 female fibromyalgia patients to 31 female healthy controls. They were analyzed for biochemical clinical parameters, for autoimmune markers, and were subjected to ^1^H-NMR metabolomics analysis. To identify a correlation between the metabolomic profile and the psychic condition, a subset of 19 fibromyalgia patients was subjected to HAM-A and HAM-D Hamilton depression tests. Multivariate statistical analysis showed the dysmetabolism of several metabolites involved in energy balance that are associated with systemic inflammatory conditions. The severity of depression worsens dysmetabolic conditions; conversely, glycine and glutamate, known for their critical role as neuromodulators, appear to be potential biomarkers of fibromyalgia and are associated with different severity depression conditions.

## 1. Introduction

Fibromyalgia syndrome (FMS) is a chronic syndrome characterized by pain, muscle stiffness, and joint and tendon pain [1,2,3]. The primary disorders of FMS are restless sleep, tiredness, fatigue, anxiety, depression, and intestinal function disorders. The areas most affected are the neck, buttocks, shoulders, arms, upper back, and chest [4,5,6].

Fibromyalgia is a gendered pathology, with higher incidence in females [7]. Therefore, fibromyalgia is one of the chronic conditions of interest in gender-specific medicine. Developed over the last two decades, gender-specific medicine is a branch of medicine that considers the influence of gender on the pathophysiology of many disorders. It focuses on the differences in pathophysiology and clinical treatment of certain diseases with a higher incidence in one gender [8]. Previous scientific evidence has identified fibromyalgia as a gender-related disease with a 9:1 incidence ratio between women and men. Fibromyalgia has a different pathognomonic profile in females than in male subjects, with higher indices in the severity of pain, overall fatigue, and psyche conditions in women than in men [9]. The causes of this gender difference are still being debated. However, they might result from the different impact that biological, psychological, and sociocultural factors have on males and females [10].

The currently known biological factors underlying the onset of fibromyalgia concern alterations in the hypothalamic–pituitary–adrenal axis. These cause alterations in the production of stress regulating hormones, such as cortisol. Serological biochemistry abnormalities show that fibromyalgia presents an autoimmune nature as characterized by the production [11] of serotonin antibody(5-HT), anti-ganglioside antibody (anti GD-3, anti GM-1, anti-GQ1b), and antiphospholipid antibody (APL) [12].

Fibromyalgia is strictly related to the psychic conditions, with disorders affecting cognitive and emotional spheres [13,14,15].Both biological and psychological dysfunction have a higher incidence in women than men, worsening their clinical conditions [11].

Depression is one of the most common mental disorders, and it is characterized by sleep disorders, interest deficit, guilt, energy deficit, and appetite deficit [16]. The diagnosis of depression is primarily carried out by excluding the presence of other neurological disorders by laboratory analysis and, subsequently, evaluating the psychic picture through structured psychiatric diagnostic interviews such as the diagnostic interview schedule (DIS) [17], as well as diagnostic tests such as the Hamilton anxiety test and the depression test (HAM-A and HAM-D). According to the HAM-A or HAM-D score, these tests evaluate an individual’s state of anxiety and depression, which are both pathological states associated with HAM-A > 17 and HAM-D > 21 [18].

Over the past decade, metabolomic studies have played a significant role in identifying the metabolomic profile that characterizes many pathologies [19,20,21,22,23]. Nuclear magnetic resonance (NMR) spectroscopy is a suitable technique for omic approaches, enabling the qualitative and quantitative detection of low-molecular-mass compounds in biological samples [24].

The metabolomic profile of fibromyalgia patients has previously been characterized using mass spectroscopy (MS) and NMR spectroscopies. Abnormal urinary concentrations of citric acid, two hydroxybutyrates, and threonine have been identified [5] with the dysregulation of numerous pathways related to energy metabolism [25].

In the present work, we studied 31 female fibromyalgia patients compared to 31 healthy female controls. They were analyzed for biochemical and clinical parameters, autoimmune markers, and the NMR metabolomics profile. The study of the complete patient set led to the identification of fibromyalgia biomarkers, thus confirming the known metabolic abnormalities [25]. To unveil a relationship between the fibromyalgia metabolic profile and the severity of depression, we analyzed the clinical and metabolomic parameters of a 19 fibromyalgia patient subset compared to the patients’ psychic conditions. Our analysis was based on the HAM-A and HAM-D Hamilton depression tests.

Our study demonstrates the alteration of metabolic pathways involved in the energy balance and associated with systemic inflammatory conditions. The severity of depression worsens dysmetabolic conditions, while glycine and glutamate, known for their critical role as neuromodulators, appear as potential biomarkers of fibromyalgia associated with different severity depression conditions.

## 2. Results

### 2.1. Clinical and Autoimmune Analysis

Blood sera of 31 female subjects diagnosed with fibromyalgia, according to the revised diagnostic protocols [26], and 31 healthy controls were analyzed for the following biochemical and clinical parameters: hemoglobin, creatinine, uric acid, parathormone (PTH), calcium, creatine kinase (CK), sideremia, ferritin, fibrinogen, C reactive protein (PCR), and rheumatoid factor (RF).

To analyze autoimmunity, fibromyalgia patients were evaluated for antinucleus antibodies (ANA), nuclear extractable antigens (ENA), and anti-parietal cell antibodies (APCA) [27,28]. A negative ANA titer was considered as ANA < 80 U/mL, while a positive titer was ANA > 160 U/mL. By following this criterion, 19.35% of the patients were ANA negative, 51.61% were weakly positive, and 9.68% were positive. All patients tested were negative for ENA and APCA, showing serum concentration values lower than 20 UC and 5 U/mL, respectively.

### 2.2. Psychological Test Results

Fibromyalgia female patients were diagnosed with depression using Hamilton anxiety (HAM-A) and Hamilton depression (HAM-D) [29,30] tests. The pathological condition was considered for HAM-D > 17 and HAM-A> 21. Accordingly, seven subjects reporting scores of HAM-A > 17 and HAM-D > 21 manifested severe depression, whereas 12 patients reporting scores of HAM-A > 17 and HAM-D < 21 manifested moderate depression.

By defining severe depression and moderate depression clusters, we calculated the variable influence on the projection (VIP) score using the R package [31]. Therefore, we identified variables discriminating between the two groups as those having a VIP value of >1. Table 1 shows that the discriminating clinical parameters for fibromyalgia patients are ferritin, reactive protein C (PCR), and creatine kinase (CK). Moderate depression fibromyalgia patients show higher ferritin and CK concentrations, whereas severe depression fibromyalgia patients show lower PCR concentrations. No significant correlation was observed between autoimmune parameters and psychological tests.

### 2.3. Statistical Analysis

#### Statistical Analysis of NMR Data

Matrices, including metabolites and their concentrations derived from ^1^H NMR 1D-NOESY spectra, were analyzed with MetaboAnalyst 4.0 using univariate and multivariate statistical analysis [32,33]. Using t-tests *p*-value < 0.05 and fold change ≥75%, D-glucose (*p*-value: 2.34 × 10^−28^), 2-hydroxybutyrate (*p*-value: 8.69 × 10^−28^), citric acid (*p*-value: 1.21 × 10^−24^), L-tryptophan (*p*-value: 4.21 × 10^−21^), and L-carnitine (*p*-value: 1.02 × 10^−16^) as criteria resulted in metabolites significantly discriminating female fibromyalgia patients in comparison to healthy female controls (see Appendix A).

An analysis of related operating characteristic (ROC) curves on the same dataset [34,35,36] demonstrated increases in 2-hydroxybutyrate, hypoxanthine, acetic acid, L-carnitine, L-proline, and L-tryptophan (100% AUC) in patient sera, with a concurrent decrease in D-glucose (see Appendix A).

A multivariate data analysis (MVA) using non-supervised (PCA) and supervised methods (PLS-DA) was carried out [37] on a matrix, including 42 metabolites collected from 62 samples (deriving from 31 fibromyalgic and 31 healthy female subjects) [38]. Normalization by constant sum and Pareto scaling was applied to the dataset. The first component explained the 47.7% variance, while the second component explained the 11.3% variance. The choice of components was made considering the confidence value (R2X) of 96% and the predictive value (Q2X) of 95% on the first component and 98–95% on the second component (see Appendix A). Figure 1a,b and Figure 2a,b show PCA and PLS-DA scores and loading scatter plots. Discrimination between the two metabolomic profiles is evident; according to the PLS-DA analysis, L-lactic acid and L-serine mainly characterize the patient group, whereas D-glucose and L-threonine characterize the healthy control group.

To evaluate the impact of the single metabolite in discriminating female fibromyalgia patients from healthy female controls, we performed a VIP score analysis (Figure 3). Accordingly, the metabolite considered as the best classifiers between the two clusters (VIP score > 1) is D-glucose; L-threonine decreased in fibromyalgia patients, while L-proline, citric acid, and 2-hydroxybutyrate increased in fibromyalgia patients. To evaluate the impact of the observed metabolite abnormalities on the biochemical pathways, we performed a metabolic pathway analysis. Table 2 reports the matched pathways, classified according to the *p*-values, the false discovery rate (FDR), and the number of hits found in the KEGG database [39].

### 2.4. NMR Data and Psychological Tests

FMS is often associated with an abnormal psychic condition typical of depression and anxiety [4,40,41]. To evaluate a possible correlation between the metabolomic and psychic profiles, we repeated the MVA by considering the results of HAM-D and HAM-A depression tests on the sub-group of 19 female fibromyalgia patients; this was performed in conjunction with the metabolomic data. Twelve out of 19 fibromyalgia patients were classified as moderately depressed, and seven were classified as severely depressed [42,43].

PCA and PLS-DA evidenced no metabolomic discrimination by considering healthy controls, moderate depression patients, and severe depression fibromyalgia patients (see Appendix A). However, ROC curve analysis indicates that glycine, betaine, glutamate, glutamine (AUC = 0.65), proline (AUC = 0.65), and creatinine (AUC = 0.65) can be considered biomarkers of depression severity in fibromyalgia patients when AUC values are given as follows: glycine (AUC = 0.69), betaine (AUC = 0.68), and L-glutamate (AUC = 0.65) (see Appendix A). At the same time, VIP score analysis (Figure 4) indicates a progressive increase in concentrations of hypoxanthine, acetic acid, creatinine, 2-hydroxybutyrate, and betaine from the healthy subjects (0) to those affected by severe depression (2). An opposite trend is evident for glucose and lactic acid.

## 3. Discussion

FMS is a chronic and systemic syndrome characterized by muscle, bone, and joint pain. It is a gender-specific condition with a 9:1 incidence ratio between women and men [9].

Although the mechanisms underlying gender differences in fibromyalgia incidence are not fully known, there is an agreement that the possible causes can be found in the specific biological, psychological, and sociocultural factors that characterize the two genders [9,10,44].

Fibromyalgia syndrome is frequently associated with mood disorders, particularly depression. To investigate biochemical correlations between fibromyalgia and depression and to identify possible common fibromyalgia and depression biomarkers, we performed a ^1^H-NMR metabolomics study on the blood sera of 31 female fibromyalgia patients; then, NMR data were correlated to the results of HAM-A and HAM-D tests from 19 out of 31 fibromyalgia patients.

MVA statistical analysis of our NMR data confirms previous evidence of the specific metabolomic pathways discriminating healthy female controls from female fibromyalgia patients (Figure 1 and Figure 2) [45]. The metabolites glucose, citric acid, and 2-hydrossybutirrate are significant disease classifiers (Figure 3 consistently with a pathological picture characterized by an alteration in energy-related biochemical pathways. This evidence is reinforced by the occurrence of gluconeogenesis and glycolysis biochemical pathways among those significantly perturbed (*p*-value = 8.12 × 10^−8^ ), as reported in Table 2.

The association between fibromyalgia and alteration of energetic metabolism could explain the significant female gender incidence of the pathology. Indeed, alteration in energy-related pathways is correlated to the hormonal estrogen pathway: previous studies have shown that 17β-estradiol has a critical role in the brain to regulate energy homeostasis, with implications on glycolysis and gluconeogenesis [46,47,48]. The alteration of glucose metabolism is in line with abnormal citric acid concentrations (AUC = 100% Appendix A) [45]. Increased concentrations of citric acid (Figure 3 and Appendix A), together with increased acetate and hypoxanthine levels—the AUCs of all these metabolites are 100%—are typical of a high incidence of anaerobic muscle metabolism [49]. This metabolic feature, also evident in the lactate transport dysmetabolism observed in Reactome analysis (*p*-value: 0.005) (Table 2), induces chronic fatigue and frequent migraines, the latter of which was an oft-cited complaint from the fibromyalgia patients [4,8,50,51,52,53,54].

The analysis of data in fibromyalgia patients classified according to depression severity indicates that metabolites involved in energy balance, such as glucose, hypoxanthine creatinine, and acetate, are proportionally altered with the severity of the depressive condition. This shows that energetic dysbalance is the cause of fibromyalgia and depression comorbidity and that a highly perturbed energetic balance may have reciprocal consequences in terms of fibromyalgia and depression symptoms [45,55,56,57] (Table 2).

Low blood glucose concentrations are a hallmark of fibromyalgia disease [45]. As shown in Figure 4, subjects with severe depression have a progressively low glucose concentration. A correlation between mood disorders and glucose dysmetabolism has been demonstrated previously [58]. Glucose is the only energy source for brain cells, and the correct function of several biochemical pathways is necessary for the following healthy glucose metabolism: (i) glycolysis and mitochondrial oxidation; (ii) metabolization of glucose in the pentose cycle, to produce NADPH needed for reactive oxygen species removal; (iii) hormone glucocorticoids, insulin, and incretin control [59,60,61]. Recently, abnormal glucose-related metabolic markers have been found in the hippocampus and frontal cortex, both brain regions that are primarily impaired in depression conditions [58].

Our data show that acetate, hypoxanthine, and creatinine are progressively high in fibromyalgic patients with severe depression (Figure 4 and Appendix A). These metabolites are also related to energetic dysmetabolism; previous evidence proved that increased purine concentration is related to mitochondrial dysfunction, increased oxidative stress conditions (Figure 3), and muscle damage [62]. Indeed, high purine concentrations can be found in FMS [63], and they are markers for severe depression, especially in females [58].

The perturbation of several amino acid biochemical pathways (Table 2) indicates amino acid dysmetabolism in fibromyalgia patients: *Alanine*, *aspartate*, and *glutamate* biochemical pathways (*p*-value 4.09 × 10^−18^); *Glycine serine* and *threonine* biochemical pathways (*p*-value 1.20 × 10^−8^). On the other hand, a reduction in L-threonine concentrations and an increase in L-proline concentrations discriminate FM patients, as shown in VIP score analysis (Figure 3). The relationship between proline concentration and FM disease has been previously demonstrated: using MS metabolomics analysis proline concentrations were shown to be proportional with Fibromyalgia Impact Scores Questionnaire (FIQ) values [64].

Alteration in the amino acid pathways further proves the relationship between fibromyalgia and depressive syndrome. A ROC curve analysis of metabolites in fibromyalgia patients classified in moderate and severe depression patients revealed progressively altered levels of glycine and glutamate (Appendix A) (Table 1). These are important neuro-mediators whose alterations have previously been demonstrated in mood disorders as being correlated to the inflammatory processes and increased inflammation markers [65]. Moreover, abnormal glutamate and glutamine levels have been shown by NMR imaging within the insula and the posterior gyrus [66,67], both brain regions that suffer significantly in depression disorders.

Consistent with the chronic inflammatory nature of fibromyalgia, it is well known that fibromyalgia patients suffer from chronic fatigue and report increased PCR serum concentrations. This is confirmed by our biochemical, clinical data [68]. Interestingly, abnormally high PCR levels have been previously found in high severe depressive conditions [69]. Moreover, our data also evidence a reduction in ferritin concentrations in subjects with severe depression (Table 1) [70,71,72], confirming previous scientific evidence on ferritin reduction in patients affected by fibromyalgia and depression.

Among other dysfunctions such as irritable bowel syndrome, chronic fatigue syndrome, and temporomandibular disorder [73], fibromyalgia is characterized by an alteration in intestinal microflora [74,75,76]. Our data show a progressive increase in acetate and 2-hydroxybutyrate (AUC = 100%, Appendix A) in fibromyalgia patients affected by severe depression. These metabolites are catabolic products of intestinal microflora [5] and are suggestive of microbiome disorders that, commonly to fibromyalgia and depressive syndrome, have a negative synergic effect on the clinical picture as a whole.

## 4. Materials and Methods

### 4.1. Partecipants and Study Design

Thirty-one (31) participants were selected from the Clinical Pathology Laboratory of DEA III Liv. Nocera-Pagani, ASL Salerno, Italy. (Table 3) Blood sera samples were collected from 31 female subjects diagnosed with fibromyalgia according to the revised diagnostic protocols the fibromyalgia diagnostic criteria [8]. The institutional ethical committee of Azienda Ospedaliera “ASL Salerno” approved the study protocol, which followed the 1964 Declaration of Helsinki and its later amendments, and all subjects gave written informed consent.

### 4.2. Autoimmune Parameter Analysis

The IIF technique was used to detect ANA using He*p*-2 cells (Euroimmun, Lübeck, Germany) [77]. The cover sheets coated with cells fixed with acetone were cut in biochips and placed on microscope slides. Serum samples were diluted to 1:100 and incubated with the cell substrate He*p*-2 × 30 m at room temperature. They were washed with PBS-Tween, followed by incubation for 30 min with anti-human goat Igg conjugated with fluorescein isothiocyanate plus propidium iodide. Finally, after the last wash, the slides were evaluated. IIF slides were subjected to automated immunofluorescence microscopy, and fluorescence models were evaluated using the Europattern software (Euroimmun) [78].

The technique used to determine ENA and ACPA antibodies is based on the principle of chemiluminescence [79] carried out by Bio-flash (Biokit, Barcelona, Spain) [80]. All samples were performed by QUANTA Flash DFS70 CIA on Bio-flash tool (Inova Diagnostics, San Diego, CA, USA) [81]. The instrument Bio-flash^®^ is an automated analyzer for immunometric tests, and its technology is based on the reading of samples in chemiluminescence. The protocol used was the standard protocol of the instrument [81]. The cut-off line was defined as 20 chemiluminescence units (CU).

### 4.3. Psychological Test: Hamilton Anxiety Test (HAM-A) and Hamilton Anxiety Depression (HAM-D)

For each fibromyalgia patient, the depressive state was calculated using the anxiety assessment scale (Ham-A) and the Hamilton depression assessment scale (Ham-D) [82]. The Ham-A scale consists of 14 points, each of which defines the extent of symptoms such as psychological stress and mental agitation, as well as somatic symptoms such as physical disorders related to anxiety [83]. The score assigned to each question is reported on a scale ranging from 0 (not present) to 4 (serious), and where a score is less than 17, we use a slight index entity in which 18 to 24 represents mild to moderate and 25 to 30 represents moderate to severe [82]. Ham-D is a questionnaire designed for adults in which the severity of depression is evaluated by testing mood, agitation, insomnia, weight loss, and somatic disorders [84]. Each element of the questionnaire is evaluated on a scale of 3 or 5 points, and then, the total score is calculated; the evaluation time is about 20 min [18]. Different levels of depression were established based on the score obtained: (i) not depressed: 0–7; (ii) mild depression: 8–13; (iii) moderate depression: 14–18; (iv) grave depression: 19–22; (v) very serious depression: >23.

### 4.4. Sample Pretreatment for NMR Analysis

NMR sample preparation and NMR spectra acquisition were performed as previously reported [19,23,24]. To obtain the blood serum, whole blood was collected into tubes not containing anticoagulant and was allowed to clot at room temperature for 30 to 120 min. After centrifugation at 12,000× *g*, the blood serum was aliquoted and stored at −80 °C in Greiner cryogenic vials before NMR spectroscopy measurements. Before being transferred to a 5 mm heavy-walled NMR tube, samples were thawed at room temperature. NMR samples were prepared by mixing 300 μL of blood serum with 200 μL of phosphate buffer, including 0.075 M Na_2_HPO_4_·7H_2_O, 4% NaN_3_, and H_2_O. Trimethylsilyl propionic-2,2,3,3-d4 acid, sodium salt (0.1% TSP in D_2_O) was used as an internal reference for the alignment and quantification of NMR signals; the mixture, homogenized by vortexing for 30 s, was transferred to a 5 mm NMR tube (Bruker NMR tubes) before analysis started [24].

### 4.5. NMR Data Acquisition

NMR experiments were carried out on a Bruker DRX600 MHz spectrometer (Bruker, Karlsruhe, Germany) equipped with a 5 mm triple-resonance z-gradient CryoProbe. TOPSPIN, version 3.0, was used for spectrometer control and data processing (Bruker Biospin, Fällanden, Switzerland) [23]. For nonfiltered biofluids, low-mass metabolites coexist with high-mass biomolecules, such as lipids, proteins, and lipoproteins; therefore, to selectively observe small-molecule components in solutions, Carr–Purcell–Meiboom–Gill (CPMG) experiments were performed. Then, 1D-1H pulse-sequence CPMG experiments comprised a spectral width of 7 kHz with 32,000 data points; water presaturation was applied over a 3.5 s relaxation delay, and we applied a spin-echo delay of 80 ms [85]. The pulse sequence used included an excitation sculpting routine for the suppression of the water signal [86]. Due to the effect of excitation sculpting on the signal height of resonances in the region close to the water resonance [87,88], the metabolites that had resonances close to this region (ascorbate, glucose, mannose, and pyroglutamate) were quantified using resonances from those metabolites in other spectral regions. A weighted Fourier transform was applied to the time domain data with a 0.5 Hz line-broadening, followed by a manual phase and baseline correction in preparation for targeted profiling analysis.

### 4.6. NMR Data Processing

NMR spectra were manually phased and baseline corrected. The quantification of serum metabolites was achieved using Chenomx NMR-Suite v8.0 (Chenomx Inc., Edmonton, Canada) [19,89]. Briefly, the Chenomx profiler was linked to the Human Metabolome Database (HMDB), containing more than 250 metabolite NMR spectral signatures encoded at different 1H spectrometer frequencies, including 600 MHz (https://hmdb.ca/, accessed on 15 May 2021). A comparison of the spectral data obtained for each serum sample with the Chenomx metabolite library resulted in a list of compounds together with their respective concentrations and based on the known concentration of the added internal reference compound TS*P*-d_4_ (5.8 mM).

### 4.7. Multivariate Analysis

All multivariate statistical analyses (PCA and PLS-DA) were made using MetaboAnalyst 4.0 (http://www.metaboanalyst.ca/, accessed on 15 May 2021) [37]. The performance of the PCA and PLS-DA model was evaluated using a cross-validation method (Q2, R2 index). A loadings plot was used to identify significant metabolites responsible for maximum separation in the PLS-DA scores plot, and these metabolites were ranked according to their variable influence on projection (VIP) scores. VIP scores were weighted sums of squares of the PLS-DA weights, which indicate the importance of the variable.

## Figures and Tables

**Figure 1 metabolites-11-00429-f001:**
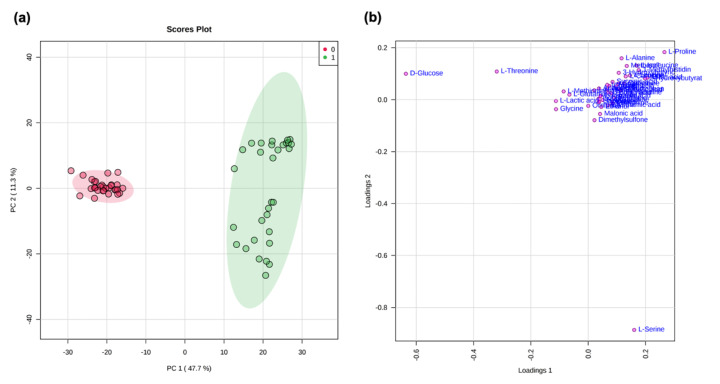
PCA score scatter plot (**a**) and PCA loading scatters plot (**b**) for the ^1^H NMR 1D-^1^H-CPMG spectra (Bruker 600 MHz) collected in the blood sera of 31 healthy female controls (0, red) and 30 female fibromyalgia patients (1, green).

**Figure 2 metabolites-11-00429-f002:**
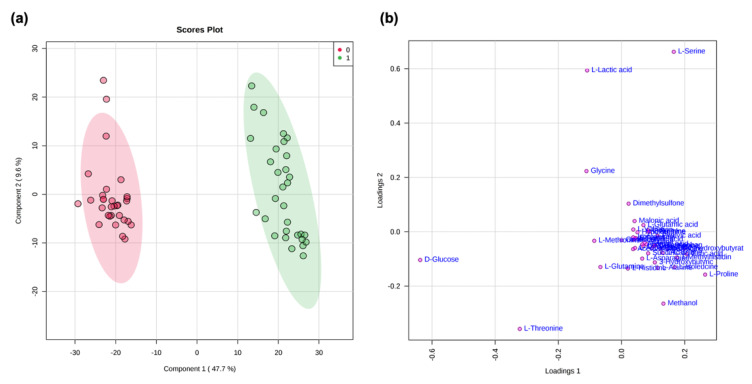
PLS-DA score scatter plot (**a**) and PLS-DA loading scatter plot (**b**) for the ^1^H NMR data collected in 1D-1H-CPMG spectra acquired at 600 MHz. Data represent the sera from 31 controls (0, red) and 30 fibromyalgic patients (1, green).

**Figure 3 metabolites-11-00429-f003:**
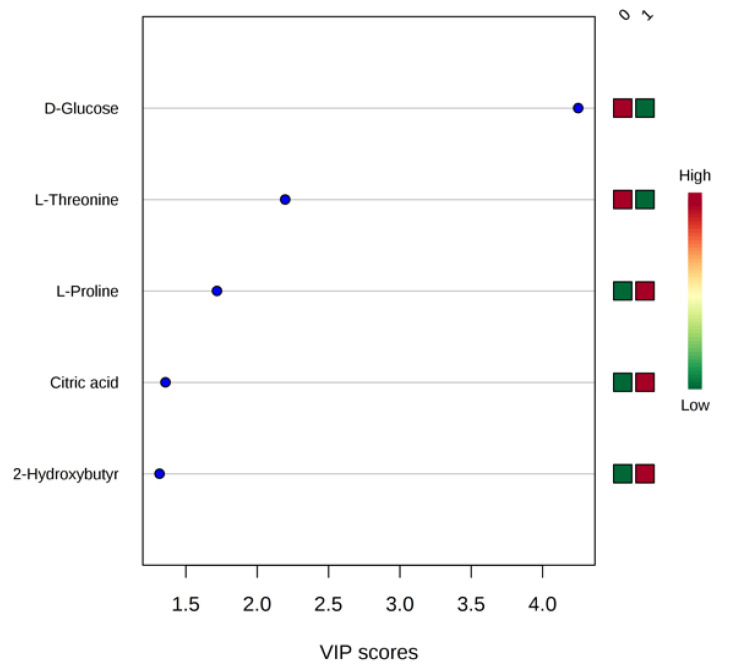
VIP score analysis: metabolites discriminating female fibromyalgia patients (1) from control (0) based on NMR metabolomic sera analysis.

**Figure 4 metabolites-11-00429-f004:**
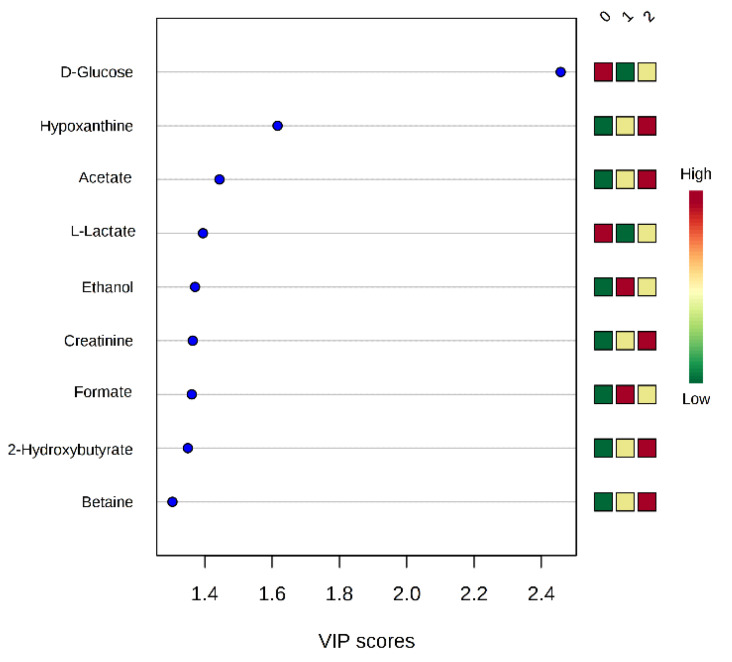
Metabolites discriminating healthy controls (0) from fibromyalgic patients with moderate depression (1) and fibromyalgia patients with severe depression (2), according to VIP score values.

**Table 1 metabolites-11-00429-t001:** VIP score and *p*-value calculated (calculated by MetaboAnalyst.R) for clinical parameters of fibromyalgia patients suffering from moderate depression (1) or severe depression (2).

	VIP Score	1	2	*p*-Value
Ferritin	2.60	+	-	0.0054
C-reactive protein	1.35	-	+	0.0134
Creatine kinase	1.00	+	-	0.0471

**Table 2 metabolites-11-00429-t002:** Pathway analysis using MetaboAnalyst and Reactome.

Pathway Name	Pathway Source	Hits	Raw *p*	FDR
Alanine, aspartate, and glutamate metabolism	Metaboanalyst 4.0	8	4.09 × 10^−18^	1.60 × 10^−16^
Purine metabolism	Metaboanalyst 4.0	2	1.42 × 10^−12^	2.04 × 10^−11^
Glyoxylate and dicarboxylate metabolism	Metaboanalyst 4.0	8	1.76 × 10^−12^	2.04 × 10^−11^
Aminoacyl-tRNA biosynthesis	Metaboanalyst 4.0	19	2.09 × 10^−12^	2.04 × 10^−11^
Glycine, serine, and threonine metabolism	Metaboanalyst 4.0	7	1.20 × 10^−8^	9.36 × 10^−8^
Glycolysis/gluconeogenesis	Metaboanalyst 4.0	3	8.12 × 10^−8^	3.52 × 10^−7^
Arginine and proline metabolism	Metaboanalyst 4.0	6	6.76 × 10^−7^	2.46 × 10^−6^
Arginine biosynthesis	Metaboanalyst 4.0	5	6.93 × 10^−7^	2.46 × 10^−6^
Pyruvate metabolism	Metaboanalyst 4.0	3	3.45 × 10^−6^	1.12 × 10^−5^
Cysteine and methionine metabolism	Metaboanalyst 4.0	3	4.70 × 10^−6^	1.41 × 10^−5^
D-glutamine and D-glutamate metabolism	Metaboanalyst 4.0	2	2.63 × 10^−5^	6.84 × 10^−5^
Nitrogen metabolism	Metaboanalyst 4.0	2	2.63 × 10^−5^	6.84 × 10^−5^
Citrate cycle (TCA cycle)	Metaboanalyst 4.0	3	4.15 × 10^−5^	1.01 × 10^−4^
Porphyrin and chlorophyll metabolism	Metaboanalyst 4.0	2	5.56 × 10^−4^	1.14 × 10^−2^
Glutathione metabolism	Metaboanalyst 4.0	3	7.58 × 10^−3^	1.48 × 10^−3^
Propanoate metabolism	Metaboanalyst 4.0	2	1.80 × 10^−3^	3.53 × 10^−2^
Valine, leucine, and isoleucine biosynthesis	Metaboanalyst 4.0	4	2.36 × 10^−1^	4.19 × 10^−1^
Tyrosine metabolism	Metaboanalyst 4.0	3	0.00010325	0.00017
Defective SLC16A1 causes symptomatic deficiency in lactate transport (SDLT)	Reactome	2	0.005466565	0.37265
Creatine metabolism	Reactome	3	0.007243495	0.37265
Proton-coupled monocarboxylate transport	Reactome	2	0.007763673	0.37265
Transport of bile salts and organic acids, metal ions, and amine compounds	Reactome	5	0.037453248	0.50662
Organic cation/anion/zwitterion transport	Reactome	3	0.038381951	0.50662
SLC-mediated transmembrane transport	Reactome	8	0.042023926	0.50662
Organic anion transporters	Reactome	2	0.042882355	0.50662

**Table 3 metabolites-11-00429-t003:** Demographic and clinical information related to fibromyalgia patients and controls.

	Fibromyalgic Group (N = 31)	Control Group (N = 31)
Sex (male/female)	0/31	0/31
Age (mean ± SD, years)	42.8 ± 14.04	50.0 ± 9.90
Number of participants psychological tests	19/31	19/31
ANA positive	19/31	0/31
ENA positive	0/31	0/31
ACPA positive	2/31	0/31
HAM-A > 17 and HAM-D < 21	12/19	0/19
HAM-A > 17 and HAM-D > 21	7/19	0/19
HAM-A < 17 and HAM-D < 21	0/19	19/19

ANA = antinuclear antibodies, ENA = extractable nuclear antigen, ACPA = anti-citrullinated protein antibodies.

## Data Availability

Data available within the article and Appendix A.

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
