# Peer review of "Fibromyalgia and Depression in Women: An 1H-NMR Metabolomic Study"

_metabolites, 2021, doi:10.3390/metabo11070429_

Round 1
Reviewer 1 Report
Marino et al. conducted a study to identify a correlation between fibromyalgia, metabolomic profile, and the psychic condition by using 1H‐NMR metabolomics analysis and depression tests. The multivariate statistical analysis identified dysmetabolism of several metabolites associated with fibromyalgia and severity of depression. Finally, they proposed several potential biomarkers of fibromyalgia associated with different severity depression conditions. Overall, the aim of the study is interesting. However, some data presentations are not very clear, and the discussion of the potential metabolomic biomarkers showed somehow disorientated. Unfortunately, the value of their proposed metabolic biomarkers is vague.
- This study only enrolled the female cases and controls. As there are different metabolic profiles between genders, the authors should state gender-associated metabolomic differences in their discussion, especially the potential hormone effects.
- As stated in the abstract and last paragraph of the introduction, glycine, and glutamate might represent potential biomarkers of fibromyalgia associated with different severity depression -conditions. However, the result section (line 175-179 ) also showed significant correlations of betaine, proline, and glutamine in the ROC curve AUC value. The authors should also discuss these findings.
- In the discussion section, the authors did not discuss their main findings of glycine and glutamate as promising biomarkers for fibromyalgia associated with depression severity.
- Is the depression also gender differences with female predominance? If yes, the authors should also take this into a statistic post-Hoc adjustment.
- Line 216-218: The authors stated higher hypoxanthine levels in patients with more severe depression and suggested progressing fibromyalgia. However, there is no further discussion about this interesting finding and no corresponding references to support this hypothesis. The authors should perform a thorough discussion about this finding.
- Line 220: I wonder if acetic acid, hypoxanthine, and creatinine are more sensitive biomarkers for fibromyalgia associated with major depression? Please clarify your findings and discuss them.
- Line 230-231: “ROC curve analysis of fibromyalgia patients clusterized in moderate depression patients and severe depression patients revealed glycine neuromediators (Figure S2).” Why did the authors not describe the predictive value of glutamate, which they mentioned in the abstract and introduction?
Reviewer 2 Report
In this study, the authors investigated the metabolic profiles of fibromyalgia syndrome (FMS) patients. Because FMS is often associated with depression, they also used the degree of depression as a factor to analyze the metabolomics data.
The experiments appear to be done appropriately. My major concern is that the strategy of data analysis without clear hypothesis; i.e., comparison looks random and not comprehensive. I also pointed out in my major comments, but Table 1 is FMS patients with moderate depression vs. those with severe depression. Fig. 1 is control vs. FMS patients. Fig. 5 (should be Fig. 4) is a different comparison. I did not understand the rationale well. As a result, this paper is failing to generate a conclusive discussion. There are a lot of errors in style, which should be thoroughly corrected in this revision.
Major comments:
1. Fig. 1A. It seems that the patients have two subgroups. Any ideas?
2. Figure 5 should be Figure 4.
3. In L104-105, there were 6 subjects of moderate depression and 6 for severe depression. In L171, there were 13 for moderate depression and 6 for severe depression. How are these numbers related? I suggest to update Table 3 including more patient information such as number of patients tested moderate/severe depression.
4. Experimental design. A 19 FMS patients subset took the depression test, and their metabolic profiles were compared with those of healthy control. The authors successfully found discriminating markers (Fig. 5) from this comparison, but it is unclear whether the difference can be attributed to FMS, depression, or the combination. What happens if the metabolic profiles are compared between FMS with no depression vs. FMS with moderate depression vs. FMS with severe depression?
5. In relation to the above comment, Figure 5 shows metabolites discriminating healthy controls from FMS patients with moderate and severe depression. How are these results comparable to previous studies on depression? Can glucose, hypoxanthine, etc. discriminate people with and without depression irrespective of FMS? I suggest to add discussion.
Minor comments:
L33: Please check the reference style throughout the manuscript. Here should be [4-6]. Also, reference comes before the period like "[1]." in this journal.
L120: This section has a different format (small fonts).
L162: Table 2 title needs to be corrected.
